# CustodyBlock: A Distributed Chain of Custody Evidence Framework

Fahad F. Alruwaili 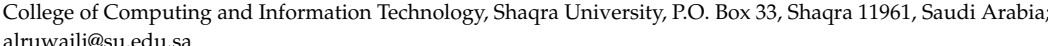

College of Computing and Information Technology, Shaqra University, P.O. Box 33, Shaqra 11961, Saudi Arabia; alruwaili@su.edu.sa

**Abstract:** With the increasing number of cybercrimes, the digital forensics team has no choice but to implement more robust and resilient evidence-handling mechanisms. The capturing of digital evidence, which is a tangible and probative piece of information that can be presented in court and used in trial, is very challenging due to its volatility and improper handling procedures. When computer systems get compromised, digital forensics comes into play to analyze, discover, extract, and preserve all relevant evidence. Therefore, it is imperative to maintain efficient evidence management to guarantee the credibility and admissibility of digital evidence in a court of law. A critical component of this process is to utilize an adequate chain of custody (CoC) approach to preserve the evidence in its original state from compromise and/or contamination. In this paper, a practical and secure CustodyBlock (CB) model using private blockchain protocol and smart contracts to support the control, transfer, analysis, and preservation monitoring is proposed. The smart contracts in CB are utilized to enhance the model automation process for better and more secure evidence preservation and handling. A further research direction in terms of implementing blockchain-based evidence management ecosystems, and the implications on other different areas, are discussed.

**Keywords:** forensics; cybersecurity; distributed ledger technology (DLT); smart contract; blockchain

## 1. Introduction

Evidence management is one of the most important problems in digital forensics. Digital proof plays a vital role in crime investigations because it is used to link persons with their criminal activities. Chain of custody (CoC) in digital forensics can be defined as a process of documenting and maintaining the chronological history of handling digital evidence [1,2]. This plays an important role in the investigation of digital forensics because it notes every detail of concern to digital evidence through different levels of hierarchy. This goes from the first responder to the higher authorities who were responsible for handling the investigation of cybercrime. Digital proof comes with its particular challenges linked with the CoC. Blockchain technology can make various sections of transactions take place. This provides massive advantages for the forensic community. In general, it is a scattered form of information that maintains tamper-proof arrangement blocks that hold a cluster of individual transactions. It executes a decentralized and fully replicated append-only ledger, present in a peer-to-peer network, originally deployed for the bitcoin cryptocurrency. All the nodes present on the chain maintain a complete local copy of the blockchain. The blockchain is an indigenous technology that has emerged for decentralized applications as the outcome of complication, privacy, and security issues present in the applications over half a century [3,4]. It is a peer-to-peer system that authorizes the users to maintain a ledger for various transactions that are reproduced, and remains identical in more than one location over multiple user servers [5].

A blockchain is essentially a block of chains, with the growing list of records referred to as blocks that are joined with cryptography [4]. Each blockchain contains a hash of a previous block, and a timestamp that keeps track of the creation and modification time of

a document. In terms of security, nobody, not even the owners of the document, ought to be able to modify it once it has been recorded (this can be done only if the integrity of time-stampers is compromised, which is rare). Blockchain was invented in 2008 and it was implemented in 2009 to function in the general public dealings ledger of the cryptocurrency (digital asset) bitcoin (cryptocurrency), a type of electronic money. Blockchain technology is decentralized (i.e., peer-to-peer). It consists of world network computers, which apply blockchain technology to put together and manage the information that records each bitcoin dealing managed by its network. The architecture of the blockchain technology, shown in Figure 1, implements a decentralized, fully replicated append-only ledger in a peer-to-peer network, originally employed for the bitcoin cryptocurrency [6].

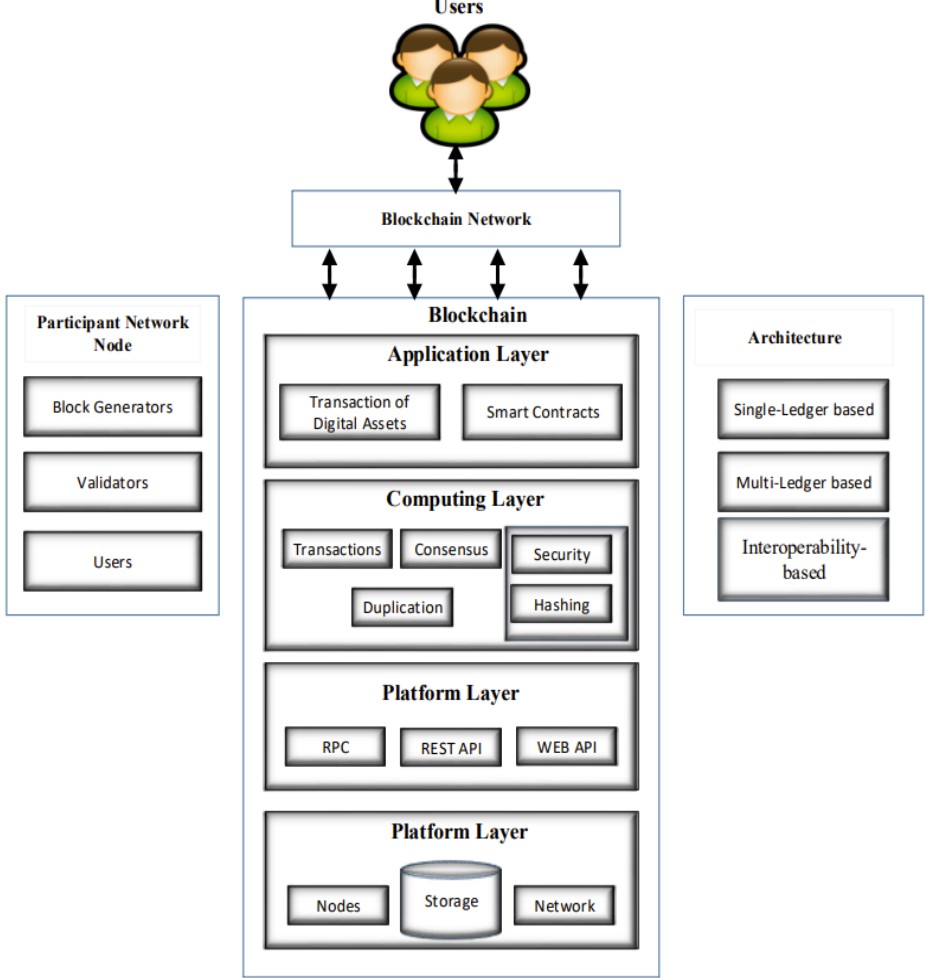

**Figure 1.** Architecture of blockchain.

Our proposed algorithm used Hyperledger blockchain technology, as it is private and open source, created by IBM. Hyperledger supports smart contract called chaincodes [7]. Hyperledger can approve blocks and send the exchange transactions to the trusted and approved blocks to be validated and approved [8]. This is the most broadly utilized blockchain stage, utilized across various ventures and use-cases. It is utilized in a few models and proof-of-concepts, and is being disseminated in record frameworks. Hyperledger is a blockchain with pluggable agreement. It is one of the more favorable projects of Hyperledger, which operates under the Linux Foundation. It is the first blockchain framework that permits the execution of dispersed applications written in standard programming. While the customary blockchain utilizes request–execute–approve engineering, Hyperledger utilizes execute–request–approve design. It utilizes a support strategy that is overseen by assigned directors and functions as a static library for exchange approval [9].

Blockchain technology provides decentralization, tracking, data transparency, security, and privacy for Internet of Things (IoT) applications [10]. IoT applications, such as medical, transportation, agriculture, etc., should be secured. Blockchain provides a high level of security and privacy for different IoT applications and transactions, and high integrity. Additionally, blockchain provides a high level of management for IoT systems by using privileged digital identities and access management [11]. Many researchers illustrate the importance of connecting the IoT systems to a blockchain-based technology to increase the security level and the IoT performance [12–14].

The production of evidence in the modern digital world is a complex task. For this reason, we consider it essential that digital evidence should be accepted as valid in court only if the chain of custody can assure exactly what was the evidence, why it was collected and analyzed, and how evidentiary data were collected, analyzed, and reported. This can be in the form of Bigdata [15,16]. Additionally, the chain of custody must demonstrate exactly where, when, and who came into contact with the electronic evidence in each stage of investigation, and any manipulation of the evidence [17,18]. The increasing complexity of forensic science in the digital area leads researchers to claim that traditional computer forensics "is on the edge of a precipice", especially because of the great diversity of electronic devices to be sized and the intensive growth of the quantity of data that must be collected and examined during a preservation in the blockchain [19,20]. This growing complexity makes it harder to create and maintain a reliable chain of custody, and exposes a wide gap between general evidentiary criteria based on traditional forensic procedures and the scientific community's point of view about the risks and conditions necessaries to consider reliable any contemporary digital evidence. This scenario circumscribes the objective of this work: to review difficulties and formulate suggestions to make a more reliable chain of custody of digital evidence, making it more consistent with court necessities regardless of country, company, or tool through which digital evidence is collected. Within this objective, this work explores gaps between the traditional chain of custody and the modern studies' point of view about the risks and requirements concerning the reliability of contemporary digital evidence.

This paper also aims to contribute to the discussion about the trend to establish a worldwide standard more suited to maintaining chains of custody throughout the lifecycle of digital evidence, and helps the improvement of new versions of the chain of custody software. This work will not define or select any standard itself, because this is a mission outside of its scope. This paper aims at addressing the challenges in the forensic evidence collection, preservation, and investigation processes, for IoT environments in the smart home domain, by exploiting the novel COC technology along with the CustodyBlock (CB) model using private blockchain protocol, and smart contracts to support the control, transfer, analysis, and preservation monitoring are proposed. The remainder of the paper is structured as follows. Section 2 presents the problem analysis and motivation of the proposed research, while the existing studies done in the area of the application of blockchain in forensics are described in Section 3. Section 4 explains the research methodology. Section 5 shows the proposed CustodyBlock model along with its architecture. Section 6 illustrates digital evidence custody (DEC). Section 7 proposes the algorithm of the proposed methodology. Section 8 gives a discussion and future work, while the conclusion and future enhancements are given in Section 9.

## 2. Problem Analysis and Motivation

In the event of a crime, the investigation process relies heavily on physical and digital evidence. The judicial system has gradually become flexible in accepting digital evidence given the fact that the handling mechanism is somewhat similar to how they treat physical evidence [17,21]. The field of digital investigation continues to grow, and therefore requires effective computer investigators with skills needed to capture the crime scene, call data records, search collected records, recover data and engage with the forensic process [7,21,22].

The issues a forensics team encounters with digital evidence are due the nature of digital information, which can be as follows:

- Easily duplicated or reproduced;
- Integrity of the evidence—altered and modified with new data or the removal of important information to the case;
- Accessibility to evidence—how and by whom is the evidence treated/managed, and what level of access control to be granted;
- Secure storage of evidence;
- Transmitted to someone else or to a different country;
- In some cases, the digital evidence is time-sensitive to the case and pre-arrest situations.

An issue arises with the gap introduced when providing digital evidence versus physical in the court of law. This is certainly becoming a challenge to the judicial system's solid and secure digital witnesses [17,23]. Another issue is that there are various practices of digital forensics activities and models. Figure 2 shows the construct in the investigation process and how physical and digital evidence support each other in creating a complete theory about the criminal case.

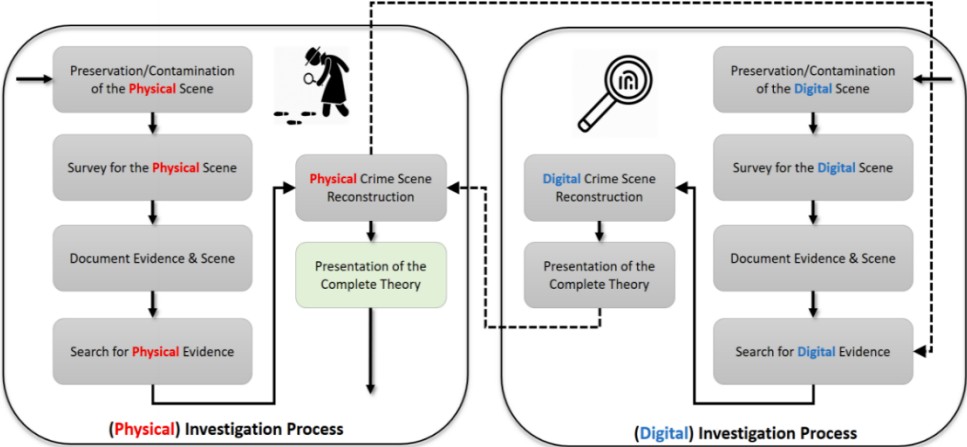

**Figure 2.** Physical and digital investigation constructs.

The paper argues for the utilization of blockchain technology to improve the chain of custody process and the evidence-handling life cycle. The blockchain is a digital ledger, system namely distributed ledger technology (DLT), and we believe it can address the current issues faced with the digital evidence handling lifecycle. DLT implementation is growing as regards its benefits in advancing sustainability, automation, and digital transformation. It also provides built-in security to control access to information and trace changes throughout the data life period. Nonetheless, the blockchain is an emerging technological advancement, hence very few concrete real-life applications are found.

The paper proposes a conceptual blockchain model, i.e., CB, to fill the gap in the literature and address the challenges faced in handling digital evidence to efficiently transform the global law practice. The CB model demonstrates how blockchain and smart contracts can provide monitored and traceable access to the evidence chain for those involved participants, e.g., regulators, courts, law firms, etc. The CB nodes are trusted network stakeholders that have a sustainable and distributed set of rules/standards. The CB smart contracts ensure the automation of rights on the evidence related to the plaintiff, defendant, and involved third party. It also has the flexibility to adapt to and comply with the applicable juridical system, and provide compliance metrics to the evidence-handling lifecycle.

Various researchers [17,20,24,25] have strongly suggested that computer forensics and law firms consider blockchain implementation to evidence handling before acquiring chain of custody services. By combining trustless, scalable, distributed, and traceable custody systems with the next generation of legal digital standards and service level agreements (SLAs), the forensics team and involved parties in a crime scene can be confident in their CoC and forensics activities. However, the focus here is on leveraging the trust, security, accessibility, provenance, and transportation of the digital evidence and handling service, the improvement of and investment in supporting law-enforcement institutions, and the investigation of cybercrime.

### 3. Existing Research

Nowadays, forensic software is used as better evidence for the process of the description and identification of the electronic user, digital signature and automatic audit trail, etc. Still, there is a great distance from the usual chain of custody software to the effective questions of the court and users. Nowadays, this process is executed by the process of CoC. The CoC is a set of consecutive documentation that records the order of custody, its control, transfer, analysis, and physical or electronic evidence. CoC contains unsafe steps during the process of investigation and at the time of submitting the evidence in court. In the past few years, various studies have been conducted in forensics based on IoT [26–30], which includes the identification of the digital evidence, its collection, storage, analysis, and distribution within IoT platform [16]. This process is entirely different from existing computer-based forensics. Since 2017, blockchain technology has been applied in various applications of digital forensics, such as the document evidencing of items, the, Soriente C [31–36]. Zhang et al. [37] proposed a provenance process model for the digital investigation using blockchain in a cloud-based environment. This is proposed to enhance the interaction trust between stakeholders present in the cloud forensics. Al-Nemrat [38] investigated the possibility of introducing blockchain-based technologies in the investigation of financial fraud in e-governance. The obtained results show that their proposed technology can be effective in determining financial fraud related to the reviews of online products.

Ulybyshev et al. [39] explained that the blockchain is used to provide the process of audibility, traceability in software development, and a role-based access control mechanism for the accesses of unauthorized data. Hossain et al. [40] proposed a forensic investigation framework based on the blockchain, which focuses on detecting various criminal incidents in the IoT and also on collecting the details of communications from different entities present in it. Their proposed framework can model the interaction of transactions. The main drawback of their methodology is the inefficiency in the collection of data and its analysis in large-scale IoT-based systems. Lone and Mir [41] proposed a digital forensic chain based on the popular blockchain platform called Ethereum. Ethereum can provide integrity, transparency and authenticity for the multiple sources of data. Various studies have been done on the process of digital investigations in the heterogeneous environment [42], security solutions in lightweight IoT devices [22], digital witnesses [31], etc. From these studies, it can be identified that the recent analysis and research on digital forensics come under two categories: focusing on assisting the law enforcement community and focusing on specific forensics applications. Cosic and Baca [43] proposed a conceptual digital evidence management framework (DEMF) to improve the chain of custody of digital evidence in all the phases of the investigation. They used a hash code for the fingerprinting of evidence, assessing hash similarity to changes control, the identification of biometrics and the authentication of digital signing, automatic and trusted timestamping, GPS, and RFID for geolocation. These components can be implemented through a database to record all the activities performed by the first responders, forensic investigators, verifier or acceptors, personnel of the law enforcement and police officers, etc. This work aims at developing a blockchain-based CustodyBlock digital forensic model that can be used in complex

cyber environments (such as IoT, cyber-physical systems, etc.) and providing an effective architecture for the proposed methodology.

## 4. Research Methodology

There are several published forensics models with detailed guidelines for the forensics team. For instance, there is a set of tasks and procedures to maintain the integrity of the captured evidence and avoid contradiction with jurisdictional laws and regulation by maintaining compliance with the applicable law requirements. The ISO/IEC 27050: 2018 is a cybersecurity catalog that highlights standards and codes of practice for electronic discovery, i.e., eDiscovery, which aims at protecting electronically stored information (ESI) including recorded data by any involved parties in the investigation process. Similarly, the digital forensic research workshop (DFRWS) model is set to protect the digital forensic process, and has six stages that start with the identification phase during an incident/event, then preservation, collection, examination, analysis, and lastly the collected evidence report is set as part of the presentation phase [17,44].

This paper utilizes a grounded and qualitative theory set by Jabareen in [17], i.e., the "conceptual framework analysis" approach to building conceptual frameworks. In addition to the current digital investigation models, e.g., ISO/IEC 27050 and DFRWS, Jabareen's approach is used to devise the CB model. This stipulates identifying a phenomenon's key factors, which when organized, clarify the proposed solution. Every key factor has its attributes, assumptions, and roles that have a specific function within the proposed model. In addition, three forms of a given assumption are presented:

1.   The way things are, i.e., ontological;
2.   How things interact/work, i.e., epistemological;
3.   The process in building the conceptual model that fits/understands the real world, i.e., methodological.

The main step in this method is to derive concepts from data within the knowledge base, i.e., literature cases, and inject the deductive procedure to generate the relationship between the model key factors. The overall process emphasizes an iterative interplay between inductive data and the analysis phases. In the end, this results in arriving at a solid examination of the potential and applicability of blockchain and smart contracts to advance CoC and the digital evidence handling model. Moreover, basic metrics [6] are considered to improve the traceability and evidence sources, such as evidence:

- Location of the data when generated;
- Type and format;
- Time elapsed since stored;
- Current control and security measures;
- Last accessibility and by who;
- Last review;
- The owner of data, who is responsible for the data;
- Transfer procedure, etc.

## 5. CustodyBlock (CB) Framework

In this section, the system features, roles, and responsibilities of the proposed CB model are defined. The CB system architecture includes participants (law enforcement, cloud service providers, validators), a consensus algorithm, smart contract, cryptographic functions, and a digital signature. The main CB model components are shown in Figure 3 and described below:

- CB Participants—The CB model ensures proper CoC documentation in order to allow the admissibility and validity of the digital evidence. This section of the model involves roles and responsibilities for those entities involved within the system. The following are major participating actors in the CB model;

- Law Enforcement (LE)—This is the major player in the CB model. The LE is a trusted third-party or government entity that is tasked with ensuring proper evidence handling procedures, e.g., collection, preservation, analysis, archiving, etc. The LE entity sets the roles for CB transactions read/write controls and ledger read/write controls. It details the rules needed to be written/coded in smart contracts to automate entity registration and onboarding, e.g., registering DW and DC;
- Digital Witness (DW)—This is the interconnected network of devices, such as laptops, smartphones, and IoTs, such as home appliances and connected vehicles, etc. DW provides collaboration on providing incident-based/sensor-based evidence within their capabilities. The captured evidence puts DC and LE forward for further investigation.

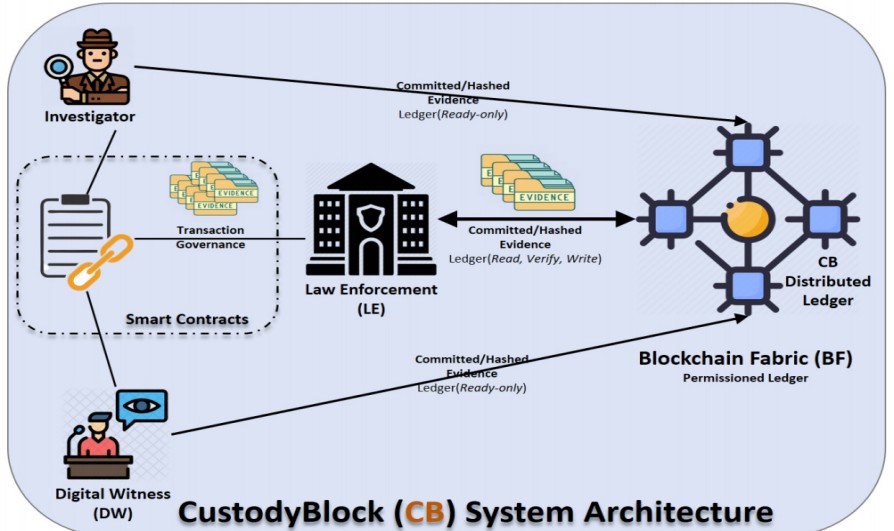

**Figure 3.** CustodyBlock framework.

## 6. Digital Evidence Custody (DEC)

An approved and accredited administrative person or entity is required to ensure the proper handling of the evidence received from DC, LE, or authorized trusted law enforcement entities. DEC can be automated through the use of smart contracts and machine learning algorithms.

Blockchain Fabric (BF): This component describes the private blockchain implementation using, for instance, Ethereum or Hyperledger fabric, as the main underlying system for CB application. Participating roles and responsibilities will act as active nodes of the CB blockchain network. The BF contains an essential element to its structure, i.e., shared ledger or DLT, which will be able to log all collective and transferred evidence and immutability shared among all the different and authorized entities. The DLT is governed by lawmakers and law enforcement institutes.

The BF has three sub-functions, which together form the operation of BF. These are:

1. Secure Transaction—This carries the evidence track records, e.g., submission, archiving, transfer, fetching, etc. Each transaction entails necessary information and a unique identifier. Information details are set as per forensic investigation standards to include data type, timestamp, submitter and receiver IDs, geographical locations, etc. The transaction is then hashed, and once verified by the consensus algorithm, will be stored in the CB DLT and distributed among all active network nodes;
2. Smart Contract—Each transaction can be automated using a smart contract. A smart contract is a set of predetermined executable instructions based on the nature of a certain transaction or input. An output can also trigger another smart contract. For example, a case is created, the smart contract logs the submitter ID and associated evidence provided by the analysis phase. Based on the analysis output, the smart contract

initiates another instance to request more evidence from the submitter or witnesses. If the submitted evidence is sufficient for the case, then the smart contract proceeds to the analysis and investigation procedures. Additional steps in the investigation process, e.g., evidence transfer and archival, are not presented in this paper;

3.  Consensus Node—This is a function with a set of rules that is responsible for maintaining, verifying and approving BF records/transactions and updating the ledger. It also ensures trustworthiness when reliability, availability, accuracy, and authenticity are built in by design. The on-chain governance of the CB blockchain is achieved by consensus nodes in not only restricting access to the CB ledger, but also who can perform different actions, e.g., validation of transactions. There are different implementations of consensus algorithms, such as proof of work (PoW), proof of stake (PoS), delegated proof of stake (DPoS), practical byzantine fault tolerance (pBFT), proof of authority (PoA), etc. A private (permissioned) implementation of the CB model is suggested with the use of practical byzantine fault tolerance (pBFT) as a consensus algorithm. The pBFT is considered for the CB model with the assumption that some of the consensus nodes may act faultily or maliciously in the network, hence our taking proactive measures to ensure consistent and valid voting/validation, which is shown in Figure 4. The pBFT does not scale to accommodate other blockchains or larger volume, but to maintain evidence handling, the author believes it should suffice.

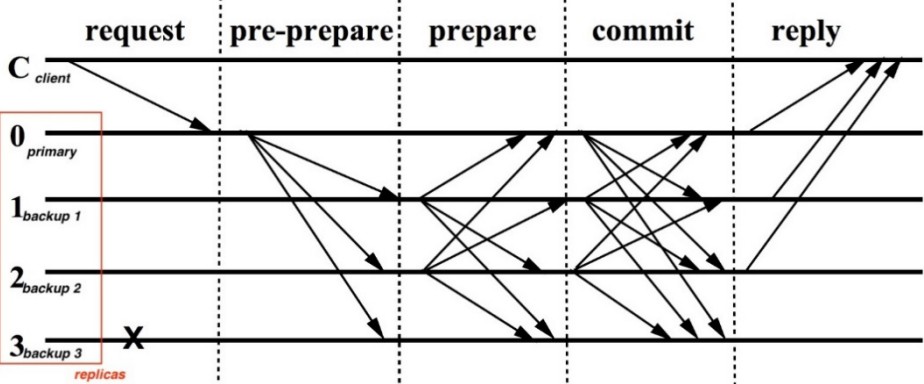

**Figure 4.** Operation phases of pBFT [44].

The given process achieves consensus if the majority of the CB network nodes agree on the same output value. The pBFT phases are:

a.  A "request" for evidence handling procedure is received;
b.  A "pre-prepare" phase to include this shared request in a proposal;
c.  A "prepare" phase is set for voting/validating and coming to an agreement;
d.  A "commit" phase allows each consensus node to communicate to each other their results, and the majority agreed-upon value will be committed into the ledger and updated in the whole CB network.

## 7. Algorithm for the Proposed Methodology

The following programing is the pseudo code of the algorithm of the proposed mechanism using python and the Hyperledger platform. Our algorithm creates a number of blocks with different hashes and with timestamps to approve, verify and track the transactions made by the blocks. The blocks' authentication uses public key infrastructure (PKI) with smart contracts to approve the creation of the created evidence. Figure 5 shows the flowchart of the proposed algorithm.

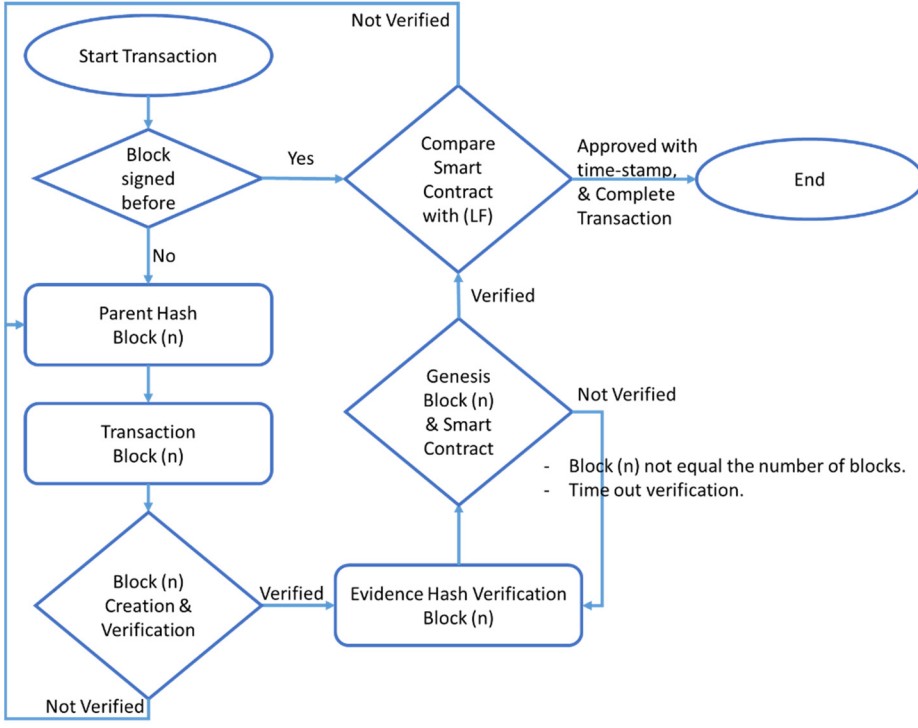

**Figure 5.** Flowchart of the proposed algorithm.

The mathematical model and algorithm are inspired by [2,45]. The following is the pseudo Algorithm 1 of our proposed algorithm which was also created in advance by the Hyperledger platform, as it is an open source code which involved the following:

---

**Algorithm 1.** Custody Block.

---

```
# (parent_hash, transactions, hash_evidence, smart_contract)
def get_parent_hash (block): return block [0]
def get_transactions (block): return block [1]
def get_hash_evidence (block): return block [15]
# function to create a block in a blockchain
"ver":1, "vin_sz":1, "vout_sz":2,
"lock_time":"Unavailable",
hash_evidence=hash ((transactions, parent_hash))
return (parent_hash, transactions, hash_evidence)
# function to create the genesis block
def create_genesis_block (transactions):
return create_block (transactions, 0)
Smart_Contract = create_genesis_block ("Create Evidence")
# function to create the smart contract block
block1 = create_block ("Smart Contract", genesis_block_hash)
# function to create the smart contract
genesis_block_hash = get_hash_evidence (smart_contract)
print "smart contract hash:", smart_contract_hash
```

---

## 8. Discussion

The CB model can be used in different applications, such as private blockchain or IoT applications. This model is a novel technique inspired by the Hyperledger technology from IBM, which uses smart contracts and public key infrastructure (PKI) to achieve confidentiality, integrity, availability, authentication, and authorization. PKI-based solutions encrypt the transaction transmitted from the authorized source peer to the destination. The

proposed model can be used to authenticate IoT users and devices to avoid different kinds of inside and outside threads and attacks.

Blockchain is a decentralization technology that stores the data and the transactions into a distributed peer-to-peer blocks. These blocks can be used to create and verify transactions using hashing and cryptographic algorithms. Zero-knowledge proof (ZKP) is a new technology to add more security to the blockchain technology by using a third party, called the "prover", who can prove and verify the value without getting any information about the transaction, but who only possesses the secret information [46–48]. Liu, in 2019, used ZKP and multi-factor authentications to secure the verifier identity, which can also be used in many IoT devices and applications. The following three properties need to be satisfied [49]:

- Completeness— knowing a spectator and the authenticity of a statement, the prover can convince the verifier;
- Soundness—a malicious prover cannot persuade the verifier in the situation that the assertion is bogus;
- Zero-information—the verifier asserts nothing aside from that the assertion is valid.

### 9. Conclusions and Future Work

In this paper, a blockchain-based protocol, and smart contracts to support the control, transfer, analysis, and preservation monitoring, have been proposed. The proposed method utilizes a conceptual blockchain model called the novel COC technology, along with the CB model using private blockchain protocol and smart contracts, to fill the gap in the literature and address the challenges faced in handling digital evidence to efficiently transform global law practice. The CB model demonstrates how blockchain and smart contracts can provide monitored and traceable access to the evidence chain for those involved participants, e.g., regulators, courts, law firms, etc. To ensure security in sharing the forensic data, the proposed mechanism ensures that all participating entities are authorized to exchange and share the data. The proposed model provides the platform on which the forensic data can be stored without any attacks, and acts as a better and more secure evidence preservation and handling methodology. Future enhancements can be the combination of intelligent-based methods along with the blockchain for providing privacy and security in electronic data. In addition, an addition of the biometric-based systems in the proposed system can offer future enhancements.

In the future, COC technology can be used for IoT smart home and healthcare applications to add security. In situations such as the COVID-19 pandemic, which affects human life. This can help the patients and the treatment team to secure the IoT medical devices, electronic healthcare records (HER), patient healthcare information (PHI), and healthcare applications. In addition, we will add a case study between the CB model proposed and zero-knowledge proof (ZKP) for smart home IoT applications with multi-factor authentication to secure the IoT user's identity.

**Funding:** This research received no external funding.

**Institutional Review Board Statement:** Not Applicable.

**Informed Consent Statement:** Not Applicable.

**Data Availability Statement:** Data available on request due to restrictions e.g., privacy or ethical.

**Conflicts of Interest:** The author declares no conflict of interest.

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
