# Peer review of "CustodyBlock: A Distributed Chain of Custody Evidence Framework"

_information, doi:10.3390/info12020088_

Round 1

Reviewer 1 Report

The presented paper describes some challenges in the forensic evidence collection and usage of Blockchain in the building the trusted custody evidence. 

The reference analysis of the modern sources is sufficient, the paper is structured and Introduction contains a short description of it.

The English language is good, but minor instructions related proof-reading are needed (i.e. "blockchain-based" instead of "blockchain based", "cyber-physical" instead of "cyber physical", etc.)

The overall impression is that it can be recommended for publishing but after moderate changes in the representation of the results achieved (and its scientific novelty). Particularly:

1) The basic information about Blockchain from Section 4 can be shortened to that particular extent that the brief information on the blockchain can be moved to the Introduction section;

2) Section 8 lacks the description of the programming language, technology used, basic flow-chart, etc.

3) It is stated that "This work aims at developing a blockchain based CustodyBlock digital forensic model that can be used in complex cyber environment (such as IoT, cyber physical systems, etc.) and providing an effective architecture of the proposed methodology." but there is nothing about IoT further. 

It is stated that "This paper aims at addressing the challenges in the forensic evidence collection, preservation and investigation process, for IoT environments in the smart home domain". but no specific IoT decisions are reviewed.

4) Paper lacks case study. 

Author Response

Hello and thank your for your time and feedback!

I have attached the updated manuscript and also see your comments addressed below:

The reference analysis of the modern sources is sufficient, the paper is structured and Introduction contains a short description of it.

We added more references 2020.

The English language is good, but minor instructions related proof-reading are needed (i.e. "blockchain-based" instead of "blockchain based", "cyber-physical" instead of "cyber physical", etc.).

We changed blockchain based to blockchain-based, and cyber physical to cyber-physical.

The overall impression is that it can be recommended for publishing but after moderate changes in the representation of the results achieved (and its scientific novelty). Particularly:

  • The basic information about Blockchain from Section 4 can be shortened to that particular extent that the brief information on the blockchain can be moved to the Introduction section.

We removed the blockchain section to the introduction section.

  • Section 8 lacks the description of the programming language, technology used, basic flow-chart, etc.

We added the programming language and technology used to the section titled “Algorithm for the Proposed Methodology”.

  • It is stated that "This work aims at developing a blockchain based CustodyBlock digital forensic model that can be used in complex cyber environment (such as IoT, cyber physical systems, etc.) and providing an effective architecture of the proposed methodology." but there is nothing about IoT further.

It is stated that "This paper aims at addressing the challenges in the forensic evidence collection, preservation and investigation process, for IoT environments in the smart home domain". but no specific IoT decisions are reviewed.

We added the following paragraph to the introduction section which illustrate how can blockchain technology added extra level of security to the different IoT applications.

Blockchain technology provides decentralization, tracking, data transparency, security, and privacy for Internet of Things (IoT) applications (Yang, 2019). IoT applications such as medical, transportation, agriculture, etc. should be secured. Blockchain provides high level of security and privacy for different IoT applications, transactions, and high integrity. Additionally, blockchain provides high management for IoT systems by using privileged digital identities and access management (Mamdouh, 2020). Many researchers illustrate the importance of connecting the IoT systems to a blockchain based technology to increase the security level and the IoT performance such as (Ernest, 2020), (Liu, 2019), and (Kim, 2019).

  • Paper lacks case study

Discussion section.

Reviewer 2 Report

The authors made some fair work but need to do extra work in order their manuscript to be more scientific and not just a report.

  1. The authors don’t provide any evidences regarding the choice of Hyperledger Fabric
  2. Moreover private transactions would need to be accompanied with anonymous client authentication mechanisms to avoid leaking the connection between the identity of the transaction’s creator and the ledger stored (hashed) data, so the authors need to adopt in their approach Zero-knowledge proof (ZKP) technology, or Privacy-preserving exchange of assets with Zero-Knowledge Asset Transfer (ZKAT), etc.
  3. A flow chart need to be provide in order the author’s approach to be better presented
  4. A math model need to be adopted for the proposed methodology
  5. A git-hub repository need to be adopted in order to present the code that the authors developed
  6. Many of the references are very stale, so the authors are strongly recommended to renew them
  7. At the conclusion section the authors need to provide their contribution to the real word.

Author Response

Hi,

Thank you for your time and feedback on my paper. I have updated the manuscript (see it attached) and also addressed all your comments below:

The authors made some fair work but need to do extra work in order their manuscript to be more scientific and not just a report.

  1. The authors don’t provide any evidences regarding the choice of Hyperledger Fabric.

We added the following paragraph to the introduction section which illustrate the importance of choosing Hyperledger.

Our proposed algorithm used Hyperledger blockchain technology as it is private and open source created by IBM. Hyperledger supports smart contract which called chaincodes (Ramezan, 2018). Hyperledger can approve blocks and sent the exchanges transactions to the trusted and approved blocks to be validated and approved (Bozic, 2016). This is the most broadly utilized blockchain stage which is utilized across various ventures and use-cases. It is utilized in a few models, proof of concepts, and underway disseminated record framework. Hyperledger is a blockchain with pluggable agreement. It is one of the favorable to jects of Hyperledger which is under the Linux Foundation. It is the first blockchain framework that permits the execution of dispersed applications written in standard programming. While the customary blockchain utilizes request execute-approve engineering, Hyperledger utilizes execute-request approve design. It utilizes a support strategy that is overseen by assigned directors and go about as a static library for exchange approval (Pavithran, 2020).

2. Moreover, private transactions would need to be accompanied with anonymous client authentication mechanisms to avoid leaking the connection between the identity of the transaction’s creator and the ledger stored (hashed) data, so the authors need to adopt in their approach Zero-knowledge proof (ZKP) technology, or Privacy-preserving exchange of assets with Zero-Knowledge Asset Transfer (ZKAT), etc.

Added ZKP in the discussion Section with many reference which can be integrated in our proposed model and compared with it in the future work.

ZKP references used in the discussion section:

  1. Li, W., Guo, H., Nejad, M., & Shen, C. C. (2020). Privacy-preserving traffic management: A blockchain and zero-knowledge proof inspired approach. IEEE Access, 8, 181733-181743.
  2. Liu, W., Wang, X., & Peng, W. (2019). Secure remote multi-factor authentication scheme based on chaotic map zero-knowledge proof for crowdsourcing internet of things. IEEE Access, 8, 8754-8767.
  3. Partala, J., Nguyen, T. H., & Pirttikangas, S. (2020). Non-Interactive Zero-Knowledge for Blockchain: A Survey. IEEE Access, 8, 227945-227961.
  4. Raikwar, M., Gligoroski, D., & Kralevska, K. (2019). SoK of used cryptography in blockchain. IEEE Access, 7, 148550-148575.
  5. Wang, D., Zhao, J., & Wang, Y. (2020). A Survey on Privacy Protection of Blockchain: The Technology and Application. IEEE Access, 8, 108766-108781.

3. A flow chart need to be provide in order the author’s approach to be better presented.

Flow chart added.

4. A math model need to be adopted for the proposed methodology.

There is no math model, we use the hyperledger platform from IBM which is open source code and we modify it to make our model.

5. A git-hub repository need to be adopted in order to present the code that the authors developed.

Hyperledger platform from IBM is an open source code. We use it and modify it to make our proposed model.

6. Many of the references are very stale, so the authors are strongly recommended to renew them.

(we added more than 10 references from 2018, 2019, and 2020) such as the following but not limited to:

  • Adam, I. Y., & Varol, C. (2020, June). Intelligence in Digital Forensics Process. In 2020 8th International Symposium on Digital Forensics and Security (ISDFS) (pp. 1-6). IEEE. 10.1109/ISDFS49300.2020.9116442
  • Aziz, B., Blackwell, C., & Islam, S. (2013). A framework for digital forensics and investigations: The goal-driven approach. International Journal of Digital Crime and Forensics (IJDCF), 5(2), 1-22. DOI: 10.4018/jdcf.2013040101.
  • Ernest, B., & Shiguang, J. (2020). Privacy Enhancement Scheme (PES) in a Blockchain-Edge Computing Environment. IEEE Access, 8, 25863-25876.
  • Mamdouh, M., Awad, A. I., Hamed, H. F., & Khalaf, A. A. (2020, April). Outlook on Security and Privacy in IoHT: Key Challenges and Future Vision. In Joint European-US Workshop on Applications of Invariance in Computer Vision (pp. 721-730). Springer, Cham.
  • Mante, R. V., & Khan, R. (2020, March). A Survey on Video-based Evidence Analysis and Digital Forensic. In 2020 Fourth International Conference on Computing Methodologies and Communication (ICCMC) (pp. 118-121). IEEE. 10.1109/ICCMC48092.2020.ICCMC-00024
  • Reedy, P. (2020). Interpol review of digital evidence 2016-2019. Forensic Science International: Synergy. https://doi.org/10.1016/j.fsisyn.2020.01.015
  • Singh, K. S., Irfan, A., & Dayal, N. (2019, November). Cyber Forensics and Comparative Analysis of Digital Forensic Investigation Frameworks. In 2019 4th International Conference on Information Systems and Computer Networks (ISCON) (pp. 584-590). IEEE. 10.1109/ISCON47742.2019.9036214
  • Zhang, X., Choo, K. K. R., & Beebe, N. L. (2019). How do I share my IoT forensic experience with the broader community? An automated knowledge sharing IoT forensic platform. IEEE Internet of Things Journal, 6(4), 6850-6861. DOI: 10.1109/JIOT.2019.2912118.

7. At the conclusion section the authors need to provide their contribution to the real word.

We added discussion section and future work to the real word as we can use our proposed algorithm due to COVID-19.

We added a lot of references as the first edition of the paper contain 32 references and the second edition contain 53 references.

Round 2

Reviewer 1 Report

I'm OK with the improvements and the new structure of the paper.

I would recommend it for the publication.

Reviewer 2 Report

The manuscript is ready to be published